



# TephraKam: Geochemical database of glass compositions in tephra and welded tuffs from the Kamchatka volcanic arc (NW Pacific)

Maxim V. Portnyagin[1,2], Vera V. Ponomareva[3], Egor A. Zelenin[4], Lilia I. Bazanova[3], Maria M. Pevzner[4], Anastasia A. Plechova[2], Aleksei N. Rogozin[3], Dieter Garbe-Schönberg[5]

[1]GEOMAR Helmholtz Centre for Ocean Research Kiel, Kiel, 24148, Germany
[2]V.I.Vernadsky Institute for Geochemistry and Analytical Chemistry, Moscow, 119991, Russia
[3]Institute of Volcanology and Seismology, Petropavlovsk-Kamchatsky, 683006, Russia
[4]Geological Institute, Moscow, 119017, Russia
[5]Institüt für Geowissenschaften, Christian-Albrecht-Universität zu Kiel, Kiel, 24118, Germany

*Correspondence to*: Maxim V. Portnyagin (mportnyagin@geomar.de)

**Abstract.** Tephra layers produced by volcanic eruptions are widely used for correlation and dating of various deposits and landforms, for synchronization of disparate paleoenvironmental archives, and for reconstruction of magma origin. Here we present our original database TephraKam, which includes chemical compositions of volcanic glass in tephra and welded tuffs from the Kamchatka volcanic arc. The database contains 7049 major element analyses obtained by electron microprobe and 738 trace element analyses obtained by laser ablation inductively coupled plasma mass spectrometry (LA-ICP-MS) on 487 samples collected in proximity of their volcanic sources in all volcanic zones in Kamchatka. The samples characterize about 300 explosive eruptions, which occurred in Kamchatka from the Pliocene until historic times. Precise or estimated ages for all samples are based on published $^{39}$Ar/$^{40}$Ar dates of rocks and $^{14}$C dates of host sediments, statistical age modelling and geologic relationships with dated units. All data in TephraKam is supported by information about source volcanoes and analytical details. Using the data, we present an overview of geochemical variations of Kamchatka volcanic glasses and discuss application of this data for precise identification of tephra layers, their source volcanoes, temporal and spatial geochemical variations of pyroclastic rocks in Kamchatka. The data files described in this paper are available on ResearchGate at https://doi.org/10.13140/RG.2.2.23627.13606 (Portnyagin et al., 2019).

## 1 Introduction

Tephra layers are widely used for correlation and dating of various deposits and landforms, for the synchronization of disparate paleoenvironmental archives, and for reconstruction of magma origin and temporal evolution. These applications are in high demand in paleoclimatology, paleoseismology, archaeology and other Quaternary science disciplines (e.g. (Lowe, 2011)), as well as in petrology and geochemistry (e.g. Cashman and Edmonds, 2019; Ponomareva et al., 2015a; Straub et al., 2015). Tephra is composed by minerals, volcanic glass (melt rapidly quenched upon eruption) and rock fragments in different proportions. A major modern approach for correlation of tephra layers between different locations is using major





and trace element composition of volcanic glass (e.g. Cashman and Edmonds, 2019; Lowe, 2011; Ponomareva et al., 2015a). The composition of volcanic glasses has been shown to vary significantly on spatial scale ranging from volcanic region to a single volcano, reflecting a large variability of thermodynamic conditions of magma storage and fractionation, and the composition of crustal and mantle sources of magmas (e.g. Bachmann and Bergantz, 2008; Cashman and Edmonds, 2019;

Frost et al., 2001; Pearce, 1996; Pearce et al., 1984; Schattel et al., 2014).

Tephra often dominates the erupted products in terms of volume, eruption frequency, and variety of compositions some of which may never occur in lava. It is particularly true for highly explosive volcanic arcs where the vast majority of the magma is erupted as tephra (e.g. Kutterolf et al., 2008). Therefore, tephra studies have a large, still only partly explored potential to trace temporal and spatial variations of magma compositions in volcanic arcs (e.g. Clift et al., 2005; Kimura et

al., 2015; Straub et al., 2004; Straub et al., 2015).

The Kamchatka Peninsula (Fig. 1) hosts more than 30 recently active large volcanic centers and a few hundred of monogenetic vents comprising the northwestern segment of the Pacific Ring of Fire. Kamchatka volcanism is highly explosive. According to some estimates, Kamchatka has the largest number of Quaternary calderas per unit of arc length in the world (Hughes and Mahood, 2008).  Kamchatka tephra layers provide chronological control for deposits and events over

large areas, both in Kamchatka and farther afield, up to Greenland and North America, which is critical for many studies (e.g. Cook et al., 2018; Hulse et al., 2011; Kozhurin et al., 2014; Mackay et al., 2016; Pendea et al., 2016; Pinegina et al., 2013; Pinegina et al., 2014; Pinegina et al., 2012; Plunkett et al., 2015; van der Bilt et al., 2017). However, geochemical characterization of Kamchatka volcanic glasses is still in a developing phase. In the Kamchatka volcanic arc, the Holocene tephrochronological framework until recently has been based mainly on direct tracing of tephra layers, bulk composition of

tephra, and bracketing radiocarbon dates (e.g. Bazanova et al., 2005; Braitseva et al., 1998; Braitseva et al., 1996; Braitseva et al., 1995; Braitseva et al., 1997; Pevzner, 2010; Pevzner et al., 1998; Pevzner et al., 2006). A significant progress towards creating geochemical database of Kamchatka tephras has been achieved in the past 10 years (Dirksen et al., 2011; Kyle et al., 2011; Plunkett et al., 2015; Ponomareva et al., 2013a; Ponomareva et al., 2013b; Ponomareva et al., 2017; Ponomareva et al., 2015b). However, the published geochemical data is mostly restricted to the Holocene and does not include data on trace

element composition of volcanic glasses.

In this paper, we present TephraKam - our original, internally consistent and so far, the most complete database of glass composition from tephras and welded tuffs of Kamchatka volcanoes covering the period from the Pliocene until present (Portnyagin et al., 2019). The data has been collected during the past 10 years and includes major element compositions obtained by electron microprobe and trace element compositions of representative samples by laser-ablation inductively

coupled plasma mass-spectrometry (LA-ICP-MS). Ages based on published radiocarbon and Ar-Ar dates as well as on the age models and stratigraphy are provided for all samples. Using this data, we present an overview of geochemical variations of Kamchatka volcanic glasses and suggest some key geochemical parameters and diagrams, which permit precise identification of tephra layers and their sources, and assessing regional geochemical variations in Kamchatka. The resulting high-resolution tephrochronological framework will help decipher the temporal and spatial complexity of archaeological





records, tectonic outbursts, volcanic impact, and environmental change for this highly dynamic area. In addition, identification of tephra layers contributes to a better understanding of regional eruptive histories, magnitudes of past eruptions, volcanic hazards, and magma origin in Kamchatka.

## 2 Volcanoes of Kamchatka and studied samples

The Kamchatka Peninsula overlies the northwestern margin of the subducting Pacific plate and is one of the most volcanically and tectonically active regions in the world (e.g. Gorbatov et al., 1997). Kamchatka hosts more than 30 large active volcanoes, 40 calderas, and hundreds of monogenetic vents grouped into two major volcanic belts running NE-SW along the peninsula: Eastern volcanic belt and Sredinny Range (SR) (Fig. 1). Eastern volcanic belt includes the Eastern volcanic front (VF), rear-arc (RA) in the southern (51-53 $^o$N) and central segments (53-55 $^o$N), and the volcanic zone of the
Central Kamchatka depression (CKD) in the north (55-57 $^o$N). Definition of VF and RA volcanoes varies in published studies. In this work, VF volcanoes are defined as those located at the closest distance to the deep-sea trench along the volcanic arc. RA volcanoes are located behind the frontal volcanoes. The current configuration of the volcanic belts is believed to have existed since c. 2.5 Ma (Avdeiko et al., 2007; Lander and Shapiro, 2007; Legler, 1977; Volynets, 1994).

The products of the continuous explosive volcanism in Kamchatka during the last 2.5 Ma are not equally presented in the
depositional record. Holocene tephra layers mantle the topography and, being interlayered with paleosol or peat horizons, form a sequence that provides a nearly continuous record of the Holocene explosive activity (e.g. Bazanova and Pevzner, 2001; Braitseva et al., 1998; Braitseva et al., 1996; Braitseva et al., 1995; Braitseva et al., 1997; Kyle et al., 2011; Pevzner, 2010; Pevzner et al., 1998; Ponomareva et al., 2015a; Ponomareva et al., 2017; Ponomareva et al., 2015b). Earlier, pre-Holocene pyroclastic products are mostly ignimbrite (pumiceous or welded tuffs), which survived through glacial stages
better than loose pyroclastics and in many cases experienced alteration  (Bindeman et al., 2019; Bindeman et al., 2010; Ponomareva et al., 2018; Seligman et al., 2014). These deposits are partly eroded by glacial processes, buried by younger deposits, and/or covered with dense vegetation, which hampers their identification.

TephraKam database provides data on volcanic glass composition from 65 volcanic centers in Kamchatka. Of these centers 43 have been active in Holocene, and the remaining 22 centers have ceased their activity prior to the Holocene time. Some
volcanic centers are individual volcanic cones (e.g. Iliinsky), calderas (e.g. Kurile Lake caldera), monogenetic lava fields (e.g. Tolbachik lava field) or monogenetic vents (cinder cones and craters) while other centers combine several volcanoes or/and calderas (e.g. Karymsky center). The latter approach was used in cases when thick local pyroclastic deposits could not have been unambiguously assigned to a certain volcano within the volcanic cluster. We refine the source vent within the volcanic center where possible (e.g. Karymsky / Polovinka caldera). Eight ignimbrite units come from unknown sources so
we use coordinates of their samples instead of vent coordinates.

We have analyzed glass from 487 samples including 11 replicate samples marked as "–rep" in the database. Overall, our samples characterize about 300 individual explosive eruptions. 298 samples come from tephra fall deposits, 187 - from





ignimbrite units (42 of them welded), and two are from lava. Our sampling covers all the Quaternary volcanic belts: 25% of the samples are from VF, 27% - from RA, 40% – from CKD, and 8% – from Sredinny Range. The coverage among the volcanic centers is not uniform: some volcanoes are characterized by only one sample while others are densely sampled and analyzed. The sampling density partly reflects the amount of large explosive eruptions from a certain volcano. The analyzed

samples span an age interval from Miocene (c. 6 Ma) to historical times (Fig. 2). About 60% of samples and data presented in this database are of the Holocene age and characterize all large explosive eruptions in Kamchatka during this time (Braitseva et al., 1995; Braitseva et al., 1997; Ponomareva et al., 2013b; Ponomareva et al., 2015b) as well as many moderate-size ones. Ages, tephra dispersal areas and volumes for most of the Holocene eruptions were published earlier (Braitseva et al., 1998; Braitseva et al., 1996; Braitseva et al., 1997; Kyle et al., 2011; Ponomareva et al., 2017; Zaretskaya et

al., 2007). Pre-Holocene record of explosive eruptions is spotty, and its representativeness decreases with increasing rock age (Fig. 2). Most of the pre-Holocene samples characterize ignimbrites associated with large (diameter 5-40 km) collapse calderas. Sixty nine of ~200 Pleistocene samples characterize 38 eruptions dated with the help of radiocarbon, Ar-Ar, or age modelling while the rest of the samples provide information on earlier unreported eruptions. Analyzed samples were collected between 1975 and 2016 by 22 contributors; the largest collections come from the authors of this paper.

## 3 Methods and database structure

### 3.1 Sample preparation

The samples have been cleaned in water to wash out clay and the finest (<5 um) fraction and dried. Fine and medium grained ash has been mounted without splitting into fractions and additional crushing. Lapilli and welded tuffs have been carefully crushed in hand mortar and then mounted. The samples were mounted in 25 mm diameter, 4 mm thick Plexiglas holders with

12 or 16 of, respectively, 3 or 2 mm cells (holes through holder). Before mounting the holders were attached to hard plate using 25 mm double sided glue tape rings for electron microscopy from Plano GmbH, which have thin but strongly adhesive glue layer and very flat surface with unevenness in the range of c. 10 μm over the entire ring. Two component epoxy resin EpoThin from Buehler has been used in the course of this study. We found this type of epoxy particularly suitable for tephra studies as the epoxy has sufficiently low viscosity to impregnate fine grained samples, has good vacuum properties, is hard,

transparent and colourless. This type of epoxy also contains analytically negligible (below analytical detection limit) amounts of most major and trace elements, except ~3 wt% chlorine. After hardening, the mounts were removed from glue tape, cleaned with ethanol and water, and then ground wet using 600-1200 grit SiC sand paper and polished by hand on stiff paper (unused punch computer card from mid-20th century) placed on hard surface using KEMET diamond pastes of 6, 3 and 1 μm grain size. Final polishing was done with 0.05 μm Buehler $Al_2O_3$ suspension in water on soft ring during 1 min.

Polishing on hard surface is crucial for preparation of very fine tephras as it provides maximal flatness of small single glass shards, ensuring high quality of analysis of small glass particles by electron probe. The samples were finally washed by brush in deionized water, dried and photographed under optical microscope.



## 3.2 Electron microprobe analysis (EMPA)

The glasses were analyzed at GEOMAR (Kiel, Germany) using JEOL JXA 8200 electron microprobe equipped with 5 wavelength dispersive spectrometers including 3 high-sensitivity ones (2 PETH and TAPH). The analytical conditions were 15 kV accelerating voltage, 6 nA current and 5 μm electron beam size for all analyses. Counting times in the latest version of

the program are 5/10 s (peak/background) for Na, 20/10s for Si, Al, Mg, Ca, P, 30/15 s for Fe, K, Ti Cl, S, 40/20 s for F and 60/20 s for Mn. The counting times have been optimized several times in the course of this study. The time record of these changes is reported in TephraKam Table 1a. The changes have not affected the data accuracy for most elements, but precision of single point analyses has been improved. Decreasing counting time for Na from 20 s to 5 s in 2010, which was aimed at minimizing loss of this element during analysis, resulted in slightly worse analytical precision but accuracy has

been improved and became less dependent on reference material for standardization.

Basaltic glass (USNM 113498/1 VG-A99) for Ti, Fe, Mg, Ca, P, rhyolitic glass (USNM 72854 VG568) for Si, Al, Na, K, scapolite (USNM R6600-1) for S and Cl, all from the Smithsonian collection of natural reference materials (Jarosewich et al., 1980), rhyolitic glass KN-18 (Mosbah et al., 1991) for F and synthetic rhodonite for Mn were used for calibration and monitoring of routine measurements. Two to three analyses of all standard glasses and scapolite were performed at the

beginning of analytical session, after every 50-60 analyses and at the end. The data reduction included on-line CITZAF correction and small drift correction for systematic deviations (if any) from the reference values obtained on standard materials. The latter correction has not exceeded a few relative percent for all elements and allowed to achieve the best possible accuracy and precision. The correction resulted in a very minor change of the mean concentrations but allowed 20-40% improvement of the analytical precision (2SD; SD is sample standard deviation) and the shape of data distribution,

making it closer to Gaussian distribution (TephraKam materials in Ponomareva et al., 2017).

The glass analyses used in this study were obtained in the period from 2009 to 2019. The summary of data for reference materials collected over this period of time is presented in TephraKam Table 1b (Portnyagin et al., 2019). The data includes results obtained on major reference glasses and minerals, which were used in calibration of quantitative microprobe measurements (USNM 72854 VG568, USNM 113498/1 VG-A99, USNM R6600-1, KN18), and also results obtained for

other reference materials analyzed as unknown in the course of our study. The latter include natural glasses USNM 111240/52 VG-2 (Jarosewich et al., 1980), Lipari obsidian, Mt. Ediza Sheep Track tephra, Laki 1783 AD tephra, Old Crow tephra (Kuehn et al., 2011) and glasses made of natural rock powders ATHO-G, BM90/21-G, GOR128-G, KL2-G, StHs60/8-G, ML3B-G (Jochum et al., 2006) and artificial glass SRM NIST-612 (Jochum et al., 2011). The data demonstrates remarkable agreement with recommended concentrations for all elements and thus excellent accuracy of our

data, which reproduce reference concentrations within the reported 2 SD in nearly all cases. The latter is also true for concentrations, which exceed significantly the concentrations in reference glasses used for calibration. This is illustrated, for example, by analyses of $Na_2O$ in NIST-SRM612 glass (13.70±0.30 wt% recommended vs. 13.73±0.40 wt% measured) and MgO in GOR128-G glass (26.00±0.30 wt% recommended vs. 25.66±0.68 wt% measured). The high quality of the data has





been also confirmed by the intercomparison of electron microprobe data for volcanic glasses between different labs (Kuehn et al., 2011, GEOMAR lab is No.12).

Precision of single point analyses depends of element concentration and analytical conditions for every element. Assuming that the reference materials used in this study were perfectly homogeneous (may be not true for natural glasses containing
microlites of minerals), the precision of single point analysis of typical rhyolite can be assessed from 2SD of the long-term mean concentrations obtained for glass USNM 72854 VG568 or Lipari obsidian. Precision of single point basaltic glass analysis can be evaluated from the data on glasses USNM 113498/1 VG-A99 or USNM 111240/52 VG-2. For more precise determination of a single point analytical precision, we provide TephraKam Table 2c (Portnyagin et al., 2019) containing electronic spreadsheet where the precision for every element is calculated based on element concentration in glass taking into
account an 11-yrs reproducibility of reference materials. Correlation of the oxide concentrations in reference materials plotted against long-term relative standard deviation (2 RSD, %) allows estimating analytical detection limits – element concentration, at which 2 RSD approach 100% (TephraKam Table 1c).

During the subsequent data reduction, we excluded analyses with the totals lower than 90 wt%, which resulted from possible unevenness of sample surface, entrapment of voids or epoxy during analysis of very small glass fragments. The latter has
been also identified by unusually high measured chlorine concentrations, resulting from entrapment of epoxy resin during analysis (see section 3.1). Analyses contaminated by occasional entrapment of crystal phases, usually microlites of plagioclase, pyroxene or Fe-Ti oxides, were mostly identified and excluded on the basis of excessive concentrations of $Al_2O_3$, CaO or FeO (and $TiO_2$), respectively, compared to the prevailing composition of glasses in every sample. Because volcanic glasses can be hydrated over time during post-magmatic interaction with meteoric or sea water or contain
significant but variable amount of $H_2O$, not completely degassed during eruption, all analyses were then normalized to anhydrous basis.

### 3.3 Laser-ablation inductively-coupled plasma mass-spectrometry (LA-ICP-MS)

Trace element analyses were performed using laser ablation LA-ICP-MS at the Institute of Geosciences, CAU Kiel, Germany. Conditions of analysis are summarized in TephraKam Table 1d (Portnyagin et al., 2019). Before 2017, analyses
were performed using a quadrupole-based (QP) ICP-MS (Agilent 7500s) and a Coherent GeoLas ArF 193 nm Excimer LA system. In situ-microsampling was done with 24-50 μm pit size and 10 Hz pulse frequency at 5-10 J cm$^{-2}$ fluence. Analyses were performed using a large volume ablation cell (ETH Zürich, Switzerland) (Fricker et al., 2011). The generated aerosol was transported with 0.75 L min$^{-1}$ He and mixed with 0.6 L min$^{-1}$ Ar prior to introduction into the ICP. The ICP-MS was operated under standard conditions at 1500W and optimized for low oxide formation (typically ThO/Th ≤ 0.4%). Ca, Ti, Si
and 30 trace elements were analyzed. The calibration was based on SRM-NIST612 glass standard (Jochum et al., 2011) and matrix corrected using ATHO-G and KL2-G glasses (Jochum et al., 2006). The measured intensities were converted to element concentrations using conventional approach (Longerich et al., 1996), with $^{43}Ca$ as the internal standard and the anhydrous normalised CaO from EMPA data. Initial data reduction was performed in Glitter software (Griffin et al., 2008).



Si and Ti concentrations obtained by LA-ICP-MS were compared to microprobe data. The analyses with deviation of LA-ICP-MS and EMPA data for these elements by more than 20% relative were rejected.

Beginning from January 2017 the analyses were obtained using a QP-ICP-MS Agilent 7900 and a Coherent GeoLas ArF 193 nm Excimer LA system operated with a fluence of 5 J cm-2, at a repetition rate of 10 Hz and a 15-24 μm ablation craters.

Analyses were performed using a modified for rapid wash-out large volume ablation cell (ETH Zürich, Switzerland) in a flow of He (0.7 L min$^{-1}$) with addition of 14 mL min$^{-1}$ H$_2$. The carrier gas was mixed with Ar (~1 L min$^{-1}$) prior to introduction to the ICP-MS. Ten major elements (Si, Ti, Al, Fe, Mn, Mg, Ca, Na, K, P) and 31 trace elements were analyzed. Analyses included 20 s background (laser-off) and 30 s signal (laser-on) measurements. Dwell time for different elements varied from 5 to 20 ms depending on their abundance. One complete measurement cycle lasted 0.607 ms and initial data reduction was

performed in Glitter software (Griffin et al., 2008) that included manual selection of integration windows and preliminary calibration. The typical integration intervals for tiny tephra shards were 6-10 s and included 10-17 cycles. The intensities corrected for background and averaged over the selected intervals were normalized to the intensity of $^{43}$Ca isotope and converted to concentrations by matching the sum of major element oxides to 100 wt% (Liu et al., 2008; Pettke et al., 2004). The calibration and correction of instrumental drift used data on ATHO-G reference glass, which was measured in duplicate

after every 18 points on unknown samples. The reference concentrations of all elements, except Na, in ATHO-G were used after (Jochum et al., 2006). Na concentrations was accepted to be 4.1 wt%, ca. 10% relative higher than reported by (Jochum et al., 2006) to reproduce Na in other reference glasses analyzed by LA-ICP-MS in this study (TephraKam Table 1d, e) and better comply with Na obtained by electron microprobe (TephraKam Table 1b). Sc concentrations were corrected for SiO+ interference using reference glasses with known Sc and variable SiO$_2$ content. The data was further filtered for inclusion of

phenocryst phases by comparison of major element concentrations with those obtained by EMPA and obvious outliers were rejected to leave only glass analyses.

During all period of data collection from 2011 to 2019, BCR2-G, KL2-G and STHS60/8-G glasses (Jochum et al., 2006) were analyzed as unknown in one series with the samples (TephraKam Table 1e). The data confirms good consistency of the entire data set and no bias related to periodic instrumental upgrades. Based on this data, the analytical precision and accuracy

are typically between ±2-8 % for 20 s long analyses but the precision might be reduced for very short analyses of tiny glass shards and for elements occurring at concentrations below 0.1 ppm.

Overall, the data obtained since 2017 using very sensitive modern instrument Agilent 7900 and after implementation of additional improvements (modified cell, addition of H$_2$ in carrier gas) is more accurate compared to earlier data for the same spot size of 24 μm. Older data obtained with 50 μm spot has comparable precision with the most recent data. However, some

former data could have been affected by entrapment of crystal phases that was impossible to identify by only using data on Ti and Si concentrations. Thus, outliers in pre-2017 data should be considered with care. Besides smaller laser beam, the post-2017 data were quantified using more efficient approach by normalizing oxides to 100%. This data is directly comparable with EMPA data for all elements except volatiles F, Cl and S. Thus, contamination of these analyses by occasional entrapment of crystal phases is excluded. The recent LA-ICP-MS data also provides accurate concentrations of



Ti, Mn and P, occurring in silicic glasses in concentrations approaching and below the detection limit of our EMP analyses (0.02-0.03 wt% for these elements).

### 3.4 Tephra ages

Knowledge of tephra ages or at least approximate age ranges is crucial for their use as marker horizons. For many tephras in our database the age estimates are available from published data (Auer et al., 2009; Bazanova and Pevzner, 2001; Bazanova et al., 2019; Bindeman et al., 2019; Bindeman et al., 2010; Braitseva et al., 1998; Braitseva et al., 1991; Braitseva et al., 1995; Churikova et al., 2015; Cook et al., 2018; Dirksen, 2009; Dirksen and Bazanova, 2009; Dirksen and Melekestsev, 1999; Florensky, 1984; Kozhurin et al., 2006; Melekestsev et al., 1992; Melekestsev et al., 1995; Pevzner, 2015; Plechova et al., 2011; Ponomareva et al., 2018; Ponomareva et al., 2013b; Ponomareva et al., 2017; Ponomareva et al., 2015b; Ponomareva, 1990; Ponomareva et al., 2006; Seligman et al., 2014; Volynets et al., 1998; Zaretskaia et al., 2001; Zaretskaya et al., 2007; Zelenin et al., 2019). In this case, we report bibliographical reference and details on the age estimates and dating technique. The majority of previously reported tephra ages were obtained by radiocarbon dating of host sediments. The ages are usually published as uncalibrated [14]C dates. In TephraKam database (Portnyagin et al., 2019), the published [14]C dates have been recalculated to calibrated ages before present (cal yr BP) with 95% error interval using the most recent IntCal13 calibration curve (Reimer et al., 2013). Some calibrated ages are based on poorly documented [14]C dates and reported as approximate ages ("~" symbol) or as an age range. Holocene tephras in Northern Kamchatka as well as some Holocene marker tephras were dated with the help of Bayesian age model combining 223 individual [14]C dates (Ponomareva et al., 2017). One Holocene tephra (KHG from Khangar volcano) was found in the Greenland ice and dated with the help of the Greenland Ice Core Chronology 2005 (GICC05, Cook et al., 2018; Rasmussen et al., 2006; Vinther et al., 2006). Twenty-one welded tuff units were dated by Ar/Ar (Bindeman et al., 2019; Bindeman et al., 2010; Seligman et al., 2014). For undated samples, the age estimates were derived from their stratigraphic relationships with the dated ones. Designated age group is provided for all samples according to the geologic time scale.

### 3.5 Database structure

The TephraKam database is provided in Excel 2016 file (.xlsx) and consists of 6 pages (TephraKam Tables 2a-f, Portnyagin et al., 2019): a) Comments, b) Volcanoes, c) Sample description, d) Major elements, e) Trace elements, and f) Discrimination diagrams. Table 2a – Comments – explains abbreviations of columns in the data tables. Table 2b – Volcanoes – contains information about volcanic centers of Kamchatka, from which volcanic glass data exists and is presented in the database. Table 2c – Sample description – includes coordinates, information of sample age, outcrop, type of material (ash, pumice, and welded tuff), collector's name and other information including data for source volcano via link to Table 2b. Table 2d – Major elements contains EMPA data on individual glass shards from samples studied and related information. Table 2e – Trace elements – contains LA-ICP-MS major and trace element data on single glass shards, information on date and conditions of LA-ICP-MS analysis, trace element concentrations normalized to mantle composition



and some element ratios for plotting the data. The tables are linked to each other so that any changes in volcano or sample description will be seen in geochemical data tables. Table 1f – Discrimination diagrams – contains sample plots and coordinates of corner points to draw compositional fields of the modern volcanic zones in Kamchatka using coordinates Nb/Y vs. La/Y and Nb/Y vs. Th/Y.

## 4 Data overview

### 4.1 Spatial and temporal variations of volcanic glass compositions

Major element data is available for all samples and comprises 7049 individual analyses. Trace elements are available for 114 samples and include 738 individual analyses. About 30% of the major element data has been already published, e.g. for Shiveluch (Ponomareva et al., 2015b); Ushkovsky (Ponomareva et al., 2013b); a part of Bezymianny and Tolbachik eruptions (Ponomareva et al., 2017). Majority of the trace element data is presented here for the first time.

An overview of the available major element data is shown in Fig. 3, a common classification diagram for island-arc rocks in coordinates $SiO_2$ vs. $K_2O$ (Gill, 1981; Le Maitre et al., 2002; Peccerillo and Taylor, 1976). In this diagram, lines dividing low-$K_2O$, medium-$K_2O$ and high-$K_2O$ compositions are drawn along typical trends of magma fractionation from basalts to rhyolites. Thus, this diagram is useful to access the extent of magma fractionation and relative enrichment in $K_2O$ of parental magma and/or source rock. Basaltic glasses are very rare in tephra from Kamchatka. The vast majority of glasses have basalic andesite, andesite, dacite and rhyolite low- to high-$K_2O$ compositions. The compositions are not uniformly distributed in $SiO_2$-$K_2O$ coordinates. Some compositions are more common; the others are rather rare. For example, dacite and rhyolite tephra glasses with $K_2O{\sim}2$ wt% and $SiO_2{=}68{-}72$ wt%, very low-$K_2O$ rhyolites and alkali rhyolites with $K_2O{>}5$ % are extremely rare or unknown in Kamchatka. In turn, medium-$K_2O$ rhyolite glasses with $K_2O{\sim}3$ wt% and $SiO_2{\sim}75$ wt% are very common and characterize many eruptions from all volcanic zones.

The compositions of glasses are grouped in Fig. 3 according to their source volcano location in Kamchatka (Fig. 3a), age (Fig. 3b), type of volcano (Fig. 3c) and type of sample (Fig. 3d). Figs. 3a-c show glasses only from tephra; Fig. 3d compares glasses from tephra and welded tuffs. Glasses from VF are represented by full range of compositions from basaltic andesites to rhyolites and belong to low-$K_2O$ and medium-$K_2O$ series. RA glasses have medium- and high-$K_2O$ compositions and overlap only marginally with VF glasses. CKD glasses have similar range of compositions with RA glasses, although medium-$K_2O$ rhyolite glasses similar to VF glasses are abundant in CKD (e.g. Shiveluch volcano). SR glasses exhibit compositional bimodality. Compositions with $SiO_2$ from 65 to 72 wt% are not known in SR. The glasses have medium-$K_2O$ (some SR rhyolites) and predominantly high-$K_2O$ compositions. The glass compositions do not exhibit clear temporal variability, suggesting similar composition of erupted magmas since at least Middle Pleistocene (Fig. 3b). The compositions of tephra glasses from complex volcanoes and calderas are similarly variable and cover all compositional range. Tephra glasses from monogenetic volcanoes tend to have either the most mafic, basaltic andesite compositions (basaltic cinder cones) or rhyolitic compositions (explosive craters). Some intermediate compositions are also known but not as abundant as





previously mentioned types (Fig. 3c). In addition to tephra glasses, the database includes glasses from welded tuffs and obsidians, which are important to characterize the oldest explosive eruptions in Kamchatka. In comparison with tephras, welded tuff glasses tend to have $SiO_2$-rich dacite and rhyolite compositions, although sample with some andesitic glasses are also present (Fig. 3d). Many welded tuffs have compositions with $K_2O>5$ wt%, which is higher than in the majority of

tephra glasses. This $K_2O$ enrichment is most likely related to secondary alteration of glass as discussed in more details in section 4.3.

Data on concentrations of trace elements in glasses adds significant information, which is highly valuable for precise identification of volcanic sources as well as for petrological and geochemical applications of this database. The data for Ti, Mn and P obtained by LA-ICP-MS is generally of higher precision in comparison to EMPA data, particularly for glasses

with concentrations of these elements below 500 ppm (0.05 wt%) approaching the detection limit of EMPA.

Trace elements provided in this database belong to different groups with contrasting geochemical properties in magmatic systems, and, therefore, provide different geochemical information. Behaviour of Sr, Ti, V, Sc, P, Zr, Hf, and heavy REE is strongly controlled by solid crystalline phases. When plagioclase (Sr), Fe-Ti oxides (Ti, V), pyroxene (Sc), apatite (P), zircon (Zr, Hf) and amphibole (heavy REE: e.g. Dy, Er, Yb) crystallize from magmas, these elements behave as "compatible

elements", and their concentrations decrease in residual melts. In contrast, elements Rb, Ba, Th, Nb, Ta, Pb, light REEs (La, Ce, Pr, Nd) behave as "incompatible elements" in most magmas of Kamchatka, because they are not concentrated in solid phases and enrich residual melt. Systematics of incompatible elements can be informative of the magma source composition and subduction related parameters, such as, for example, the distance from volcano to subducting plate (e.g. Volynets et al., 1994; Churikova et al., 2001; Duggen et al., 2007). The ratios between incompatible elements do not change as magma

fractionates and are instructive to identify source volcano of variably fractionated melts. This information is quite unique in comparison to the systematics of major elements, which, in contrast to incompatible trace elements, is largely related to the conditions of magma storage and also to syneruptive crystallization and magma mixing (Cashman and Edmonds, 2019; Ponomareva et al., 2015a).

Although detailed evaluation of trace element systematics in Kamchatka glasses is beyond the scope of this work, we

illustrate in Fig. 4 some regularity in trace element composition of tephra glasses from different volcanic zones in Kamchatka, which can help identify source volcano or at least volcanic zone for tephra of unknown provenance. In this diagram, we show only Holocene and Late Pleistocene samples as their source volcanoes are reliably constrained, and the available data is the most representative compared to older volcanic rocks.

$(Nb/Y)_N$ and $(La/Yb)_N$ ratios in glasses (N denotes values normalized to primitive mantle after McDonough and Sun (1995))

reflect source enrichment in highly incompatible elements (La, Nb) relative to less incompatible elements (Yb, Y) (Pearce et al., 1995; Pearce et al., 1984). These ratios are also strongly influenced by amphibole crystallization in evolved magmas, which is important host for heavy REE and Y, but not for La and Nb (Brophy, 2008). Tephra glasses from frontal Kamchatka volcanoes have relatively low $(Nb/Y)_N <1.3$ and $(La/Yb)_N<5$. This is distinctive compositional feature of VF tephra in comparison with glass compositions from the other volcanic zones in Kamchatka. RA glasses have $(Nb/Y)_N =1.1$-





3.6 and $(La/Yb)_N$ = 3.1-10.9. Glasses from SR tephra have even higher $(Nb/Y)_N$ >4.5 and $(La/Yb)_N$ = 5.5-11.8 mostly overlapping with RA compositions. Both ratios increase with increasing the distance from the deep-sea trench. CKD glasses have $(Nb/Y)_N$ similar to those in RA glasses and $(La/Yb)_N$ >3 overlapping with RA and SR glasses.

$(Ba/Th)_N$ and $(Th/La)_N$ ratios are informative of magma source composition and related, respectively, to contributions from
slab fluid and sediment melts in source of magmas (e.g. Elliott et al., 1997; Pearce et al., 1995; Plank, 2005). Both ratios exhibit significant variations along volcanic arc and range from mantle-like values ~1 up to 4-6 times higher than in primitive mantle. On regional scale, VF tephras tend to have higher $(Ba/Th)_N$ and lower $(Th/La)_N$ than RA tephra: $(Ba/Th)_N$ =1.5-6.2 and 1.2-3.3, $(Th/La)_N$ = 0.8-2.7 and 1.0-4.4 in VF and RA, respectively). At given distance along volcanic arc, VF tephras always have higher $(Ba/Th)_N$ and lower $(Th/La)_N$ in comparison with RA tephras. SR tephra have a relatively low
$(Ba/Th)_N$ <2 and high $(Th/La)_N$ >2 in the range of RA tephra. CKD tephra has these ratios similar to VF tephra.

To sum up the overview of geochemical data, compositions of glass in Kamchatka tephra are very variable, enabling effective correlation of tephra layers as well as identification of source volcano and volcanic zone, from which unknown tephra could come from.

## 4.2 Using composition of glasses for fingerprinting ash layers in Kamchatka

TephraKam was initially created for tephrochronological needs to enable reliable identification and dating of tephra layers in Kamchatka and neighbouring areas and for identification of their sources. The data has been used in a number of publications (Cook et al., 2018; Derkachev et al., 2019; Plunkett et al., 2015; Ponomareva et al., 2018; Ponomareva et al., 2013a; Ponomareva et al., 2013b; Ponomareva et al., 2017; Ponomareva et al., 2015b; Zelenin et al., 2019). Our experience showed that ash layers produced by the largest explosive eruptions in Kamchatka can be recognized using major element
systematics of tephra glasses. Diagram of $SiO_2$ vs. $K_2O$ is useful for primary identification (Fig. 3), because it utilizes elements of contrasting geochemical properties, reflecting mantle or crustal source enrichment in incompatible elements ($K_2O$) and extent of magma fractionation ($SiO_2$). In this respect, the diagram is more informative compared to other diagrams widely used in tephrochronology such as FeO vs. CaO or FeO vs $TiO_2$, utilizing elements whose concentrations are largely controlled by crystallization processes and strongly correlate with each other. More detailed discrimination of ash
layers requires additional geochemical constraints. For example, low-$K_2O$ tephra glasses from Ksudach and Avachinsky volcanoes are well distinguished using CaO vs. $SiO_2$ systematics; medium-$K_2O$ glass from Bezymianny volcano tephra has lower $Na_2O$ compared to Shiveluch glasses; high-$K_2O$ glass from Ushkovsky tephra has distinctively higher $P_2O_5$ compared to glass from high-$K_2O$ basaltic tephra SH#28 from Shiveluch (Ponomareva et al., 2017).

In rare cases, tephra glasses from different volcanoes have hardly distinguishable major element composition. In this case
minor elements determined by EMPA (P, Cl) and trace elements by LA-ICP-MS are useful to identify source volcanoes. In Fig. 5, we illustrate this case using compositions of tephra glasses from Opala volcano (eruption OP 1356 BP) and Khangar volcano (eruption KHG6600 7490 BP). Although these tephras have very different ages, this comparison is instructive to illustrate the value of minor and trace element data to distinguish compositionally close tephras. The difference in major





elements is very subtle and mostly within long-term analytical uncertainty: Khangar glass has about 0.5 wt% lower $Al_2O_3$, ≤0.2 wt% higher CaO, ≤0.5 wt% higher $K_2O$ at given $SiO_2$ and otherwise completely overlapping $TiO_2$, FeO, CaO and $Na_2O$ contents (Fig. 5 a-c). The two tephras have however clearly different Cl content (Fig. 5d) and very different shapes of normalized trace elements (Fig. 5e), enabling clear discrimination between source volcanoes. Summarizing our experience to date, we were not able to identify any identical tephras from different volcanic sources in Kamchatka.

Distinguishing tephra layers from the same volcano is a more difficult task. However, there is a number of examples from Kamchatkan volcanoes when tephra compositions are different even on short intervals of time (e.g. Kyle et al., 2011; Ponomareva et al., 2013b; Ponomareva et al., 2017; Ponomareva et al., 2015b). For example, Ponomareva et al. (2015b) showed that even compositionally similar tephra layers of frequently erupting Shiveluch volcano can be distinguished using major element systematics in tephra glasses, particularly when the time period of eruption can be narrowed using marker layers from other volcanoes. The cases of compositionally identical products of different eruptions from the same volcanic center are also known. For example, Derkachev et al. (2019) reported two late Pleistocene layers produced by large eruptions from Gorely caldera, which have hardly distinguishable major and trace element composition of glass. On a longer time scale of hundred thousand years, the products of the Gorely caldera eruptions are more variable (Seligman et al., 2014), enabling their identification using glass composition in tephra and welded tuffs.

### 4.3 Effects of alteration on major and trace elements in glass from welded tuffs

TephraKam contains abundant data for glasses in welded tuffs of Miocene to Pleistocene age from different parts in Kamchatka. These rocks clearly represent products of large caldera-forming eruptions during Kamchatka history. Identification of their sources, age and ash distribution is of great interest. However, some welded tuff glasses in the database have signs of secondary alteration that hampers their direct interpretation as compositions representative for native glass – quenched melt. Typically, the alteration results in characteristic "spaghetti"-like textures, precipitates of tiny magnetite crystals in glassy matrix followed by complete glass replacement by microcrystalline aggregate, and development of concentric perlite texture (Fig. 6). The process of devitrification is also associated with chemical modification of welded tuffs. Spot analyses usually reveal a large and correlated variability of alkalis and alkali earth elements within single sample, which typically is not observed in volcanic glasses from pumice fragments or ash layers. Representative trace element composition of variably altered glasses from the same unit of Karymsky / Stena-Sobolinaya caldera welded tuffs is shown in Fig. 7. A major feature of the glass alteration is enrichment in $K_2O$, Rb, Li (all are monovalent alkaline elements) that is inversely correlated with depletion in $Na_2O$, CaO, Sr. Elements Ba, U, Th, Pb, Nb, Ta, Zr, Hf, Ti, P, Y, and REEs exhibit small variability and are relatively immobile during alteration. Concentrations of the immobile elements are informative of the initial concentrations in glass and can be used for correlations between different ignimbrite units and with pristine tephra glasses. An example of geochemical fingerprinting of completely devitrified samples from Pauzhetka caldera in South Kamchatka is provided by Ponomareva et al. (2018). These authors noted that elements B, Ba, Eu, and V also reveal mobility in strongly altered tuffs and should be interpreted with caution.



## 4.4 Discrimination of glasses from different volcanic zones in Kamchatka using immobile trace elements

In comparison with major elements, concentrations of some trace elements in Kamchatka glasses are more variable and exhibit characteristic regional distribution (Fig. 4). This makes it possible to use trace elements for identification of volcanic zones - sources of distal tephra. Ideally, these criteria should use immobile elements, which are unaffected by alteration of

glasses in welded tuffs and ancient tephras buried in marine sediments and other deposits. We performed a search for the most effective criteria based on trace element concentrations in glasses from this database. Based on this search, diagrams using trace element ratios Nb/Y, Nb/Y and Th/Y provide the most robust discrimination of glass compositions from different volcanic zones in Kamchatka (Fig. 8, TephraKam Table 2f). The fields of different volcanic zones are initially drawn using data on glasses from robustly identified sources of the Holocene and Late Pleistocene ages (Fig. 8 a, b). Glasses from

volcanic front, rear-arc and Sredinny Range form separate fields in these diagrams, which should be related to systematically changing conditions of magma generation with increasing distance from volcano to deep-sea trench and to subducting plate. Glasses from the Central Kamchatka Depression volcanoes largely overlap with rear-arc glasses and partly with volcanic front samples that is in agreement with geodynamic position of CKD (Fig. 1). Figure 8 c, d shows that glasses of middle and early Pleistocene age have compositions falling closely within the corresponding fields of different volcanic zones. Although

Th/Y and La/Y are somewhat scattered at given Nb/Y, the volcanic zones can be still precisely identified in most cases. This consistency of compositions suggests that Kamchatka has not been affected by major tectonic reorganization at least during Pleistocene and Holocene (c. 2.5 Ma), and the present position of ancient volcanic centres corresponds closely to their initial position. Older, Miocene-Pliocene samples are relatively rare in this database (Figure 8 e-f). However, we notice that glasses in Neogene rocks from Sredinny Range have distinctive compositions that plot outside the range of Pleistocene-Holocene SR

compositions and are more similar to modern rear-arc and volcanic front samples. This might indicate that conditions of magma generation under Sredinny Range during the Pliocene were more similar to the present-day volcanic front and rear-arc, and that Sredinny Range volcanoes were located closer to the deep-sea trench and subducting plate at that time. This conclusion is in agreement with the proposed major tectonic reorganization in Kamchatka in the Neogene and shifting of volcanic front from the Sredinny Range to its present position in Eastern Kamchatka (Avdeiko et al., 2007; Lander and

Shapiro, 2007; Legler, 1977; Volynets, 1994). Thus, the diagrams in Figure 8 can be successfully used to identify tephra erupted from different volcanic zones during the Holocene and Pleistocene. The proposed criteria are likely not valid for the Neogene time, when the Kamchatka subduction zone had different configuration. The diagrams can be particularly useful to identify the provenance of distal tephras in marine sediments offshore Kamchatka, for analysis of synchronicity of activity in different zones, and for analysis of temporal geochemical variability of volcanism.

## 5 Data availability

The archive .zip file containing tables of this data set is available on ResearchGate: https://doi.org/10.13140/RG.2.2.23627.13606 (last access: 21 October 2019) (Portnyagin et al., 2019). The authors leave

rights to update this database (correct misprints, add new data, refine ages etc.). The initially released version of TephraKam is 2019.10.21 (the number refers to the date of release in format YYYY.MM.DD). The latest version can be downloaded from ResearchGate or requested directly from the two first authors of this manuscript.

## 6 Conclusions

5 TephraKam is the largest and most comprehensive collection of internally consistent high quality chemical analyses of major and trace elements in glasses of pyroclasic rocks of Kamchatka volcanoes. Precise or estimated age is provided for every sample. Use of this database opens possibility for reliable identification and correlation of tephra layers in Kamchatka and neighbouring areas, enables dating of sedimentary archives on- and offshore Kamchatka as well as allows the multi-component petrological and geochemical analysis of composition and origin of magmatic melts, preserved as quenched glass

10 in tephra. The latter application is straightforward for rhyolite glasses, which have been shown to preserve the composition of magmas at depths (except for volatiles) and thus are informative of magma composition and its storage conditions at depth in Kamchatka (Ponomareva et al., 2015a). The amount of presented data is comparable and exceeds that available from published sources on the composition of volcanic rocks in Kamchatka (e.g. GEOROC database). For silicic compositions, this database is a major source of information.

**Author contributions.** MP and VP designed the study and wrote the paper. MP, VP, LB, MP, and AR conducted the fieldwork and collected most of the samples. MP developed protocols of the EMP and LA-ICP-MS analyses and designed the database structure. MP, VP, AP, and AR produced the EMP and LA-ICP-MS data. EZ compiled all the age data. DGS managed the laser-ablation ICP-MS analyses. All authors discussed the results and participated in preparation of the paper.

**Competing interests.** The authors declare that they have no conflict of interest.

**Acknowledgements.** The authors cordially thank all colleagues who donated their samples for this research. The field efforts of I.N. Bindeman, T.G. Churikova, O.V. Dirksen, N.V. Gorbach, B.N. Gordeichik, S.A. Khubunaya, N.A. Kim, S.P.

25 Krasheninnikov, P.R. Kyle, I.V. Melekestsev, N.L. Mironov, A.B. Perepelov, P. Rinkleff, and O.B. Seliangin are highly appreciated. We are deeply grateful to late O.A. Braitseva and V.L. Leonov for providing their samples and for long-term support of our research. We appreciate the Kronotsky Reserve help in acquiring the samples from Uzon caldera and Kronotsky Lake. We thank Mario Thöner and Ulrike Westernströer for all their efforts in managing laboratory work and many-year excellent assistance with electron probe and LA-ICP-MS.



**Financial support.** This research was supported by the Russian Science Foundation grant #16-17-10035 (partial funding of fieldwork, data processing, work on the database and manuscript). All laboratory costs related to electron microprobe and LA-ICP-MS analyses were covered by the GEOMAR Helmholtz Centre for Ocean Research (Kiel, Germany).

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



**Figure 1 Volcanoes and samples presented in TephraKam. Large red circles with labels – volcanic centers; yellow circles – sample locations. Note that Kamchatka hosts more presently inactive volcanic centers than shown on the map, but tephra samples from these extinct volcanoes were not available for this study. The background image is drawn by the authors using public domain datasets SRTM for landmass (Farr et al., 2007) and GEBCO for ocean floor (Smith, Sandwell, 1997).**

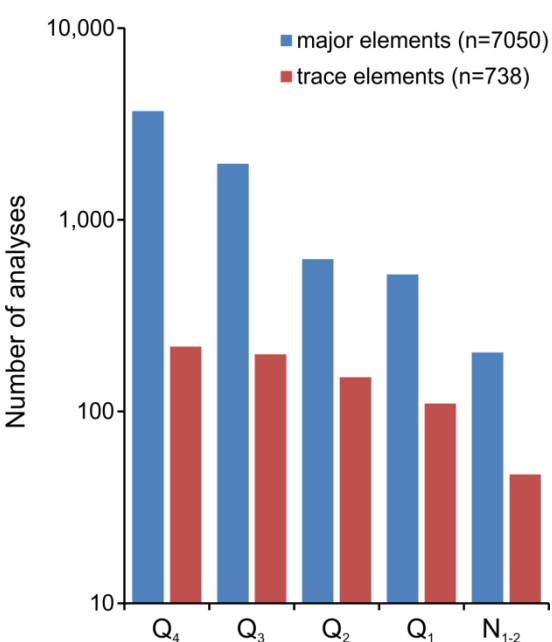

**Figure 2 Number of single spot major and trace element analyses of glass from tephra and welded tuffs of different age groups included into the TephraKam database. Note logarithmic vertical scale.**



**Figure 3 SiO$_2$ – K$_2$O variations in glasses. The glasses are grouped according to volcanic zone (a), age (b), type of volcano (c), and rock type (d). Dashed lines divide fields of low-K$_2$O (LK), medium-K$_2$O (MK), and high-K$_2$O (HK) basalts (B), basaltic andesites (BA), andesites (A), dacites (D) and rhyolites (R) after Le Maitre et al. (2002). The line dividing fields of dacite and rhyolite is drawn for the case of Na$_2$O content of 5 wt%, which is typical for high-silica glasses from Kamchatka.**

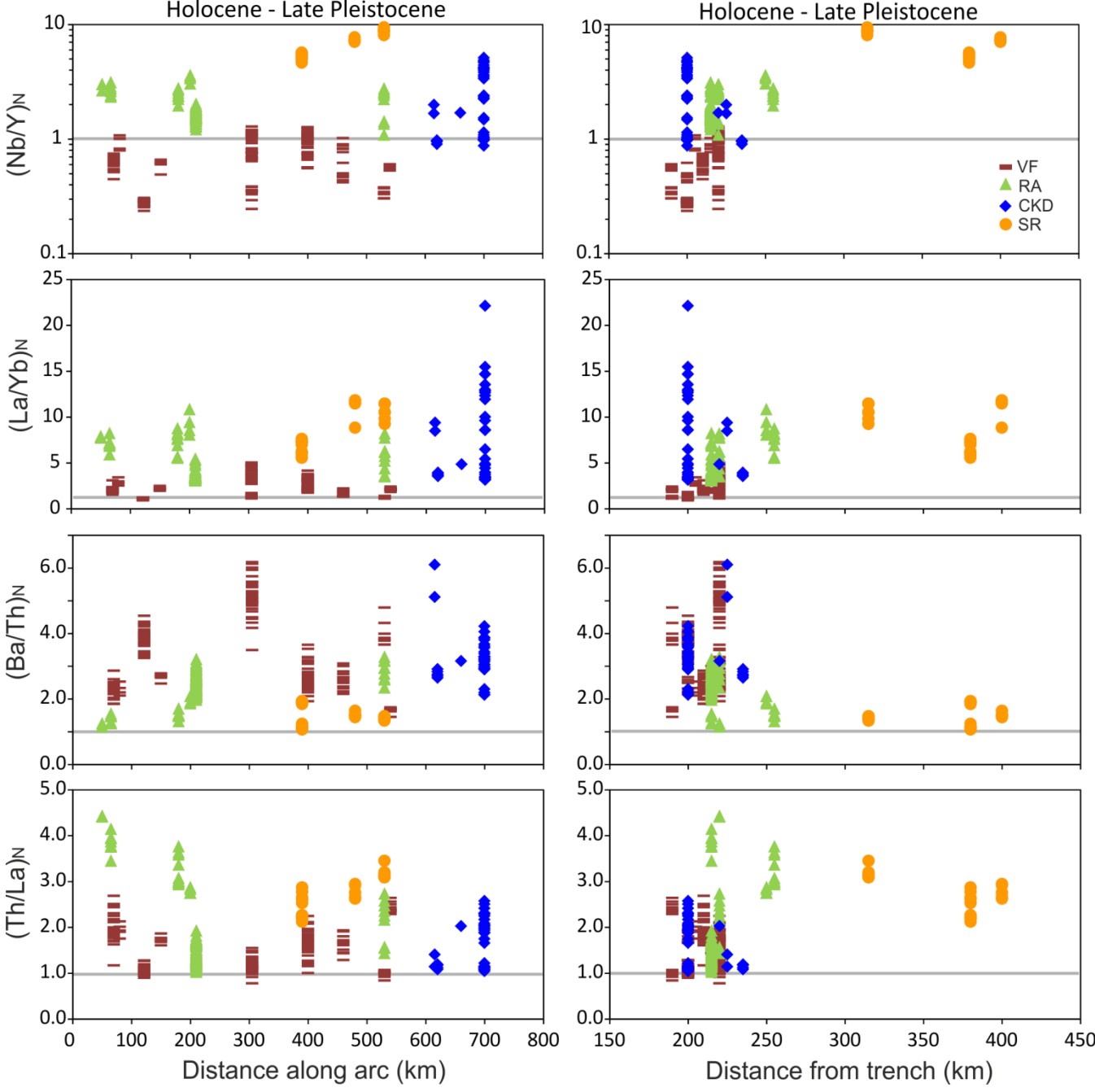

**Figure 4 Trace element variations in Holocene-late Pleistocene tephra glasses south to north (left) and across (right) the Kamchatka volcanic belts. Trace element ratios are normalized to primitive mantle values (McDonough and Sun, 1995). Nb/Y and La/Yb ratios reflect mantle source depletion/enrichment (e.g. Pearce et al., 1995) and also the extent of amphibole fractionation (e.g. Brophy, 2008); Ba/Th is a function of slab-derived fluid contribution to the source of magmas; Th/La is related to the amount of subducted sediments involved in magma generation (e.g. Elliott et al., 1997; Pearce et al., 1995; Plank, 2005).**



**Figure 5 Example of using minor and trace element data to precisely identify source volcanoes for glasses with very similar major element composition: the case of tephras from Opala (eruption OP) and Khangar volcanoes (KHG6600). Uncertainty of single points corresponds to 2 standard deviation (2s) as calculated for average KHG6600 composition using TephraKam Table 1c.**

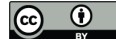

**Figure 6** Back-scattered electron images of glass devitrification in welded tuffs: a) Slight alteration along welded glass particles (Sample 198-75, Karymsky/Stena-Sobolinaya caldera); b) More advanced alteration, precipitation of magnetite (sample 169-75, Karymsky/Stena-Sobolinaya caldera); c) Strong devitrification, developed perlite texture (sample 1989L-97b, unknown source/Alney-Chashakondzha?); d) Complete devitrification (sample PAU-8; Pauzhetka caldera, no glass preserved).

**Figure 7 Illustration of chemical effects of secondary alteration on major and trace element composition of glass from welded tuffs: a) Covariations of mobile elements; b) Trace elements normalized to primitive mantle composition (McDonough and Sun, 1995). Arrows denote effect of alteration. Samples 169-75 (less altered) and 202-75 (more altered) are from Stena-Sobolinaya caldera and likely belong to the same unit judging from very close concentrations of immobile elements.**



**Figure 8 Discrimination diagrams for different volcanic zones in Kamchatka. The fields are drawn based on Holocene-late Pleistocene compositions. Coloured symbols show compositions of glasses according to their estimated ages and present-day location: Holocene-late Pleistocene (a, b), middle –early Pleistocene (c-d), and Miocene-Pliocene (e-f).**