# Peer review of "TephraKam: Geochemical database of glass compositions in tephra and welded tuffs from the Kamchatka volcanic arc (NW Pacific)"

_Earth System Science Data, 2019_

## Referee Comment (RC1) · Bärbel Sarbas (Referee) · 2 Dec 2019

The term database refers to sets of related data. In the case of TephraKam, there are four individual tables (volcanoes, samples, major elements and trace elements) which are not related. Thus, the comprehensive and unique data collection should not be called database but dataset.

---

## Author Comment (AC1) · 2 Dec 2019

Thanks for this comment. 1) According to Wikipedia, "A database is an organized collection of data, generally stored and accessed electronically from a computer system." (https://en.wikipedia.org/wiki/Database). Our data collection satisfies this definition of a database. Moreover, the four tables in the xlsx. file (volcanoes, samples, major elements and trace elements) are actually linked (related) using unique sample number and/or volcano name. In other words, if one changes, for example, the age of sample in "Samples" spreadsheet, the age will be authomatically changed in tables with major

and trace elements. All interlinked cells are identified in the database structure description. 2) On the other hand, this is correct that our data collection is not constructed as typical relation database. Our goal was to make the files as friendly for potential users as possible. Therefore, we provided the tables in MS Excel format, not in special format of database or on web. Depending on the editor's decision, we can change description of this data collection to as "dataset"; this is not very important for us "database" or "dataset".

---

## Referee Comment (RC2) · Anonymous Referee #2 · 13 Jan 2020

I believe that this is a paper that is worthy of publication. It is the culmination of a thorough analysis of many of the tephra and weld tuff deposits produced by volcanoes in Kamchatka. This original data will prove invaluable to researchers working in the region and also on a wider global scale, where ultradistal tephras are now being found. The geochemical data is of a high quality and detailed explanations are provided on the analytical conditions of the instruments used to produce the results.

The issues I have with this paper that need to be addressed are as follows and I have also highlighted them in the attached PDF file.

1. There is no mention of Miocene samples in the Abstract. A minor point, but needs correcting.

2. The discussion on tephra sample preparation for analysis on page 4 needs to include some reference to other studies. This has been widely discussed in the literature, but there is no reference to any other sources. Whilst the preparation of samples is thorough, it is not necessarily innovative and some acknowledgement of others is required here.

3. Page 4 discusses electron microprobe analysis and again there is no inclusion of any citations to other published work on this subject, other than some of the glass standards. This subject has been discussed in detail in the literature and some inclusion of this is needed.

4. Page 6 onwards discusses LA-ICP-MS analyses and there is more inclusion of published work on this subject here, although I would expect to see more.

5. I am not suggesting that this is done for the publication of this paper, but it would be good to see this data housed in a proper relational database. This would allow more complex searching than is currently possible with the Excel spreadsheets. Some mention of this as at least a possibility would be good.

6. Although not badly written, there are a lot of minor English issues with the text. I have tried to highlight these in the PDF file (Supplement) and I hope the authors find this useful. I do appreciate the difficulties in writing a paper in another language and I feel that they have done a good job, but the quality of the writing needs to be more consistent. I would hope the editors/publishers would also contribute to this.

Please also note the supplement to this comment:
https://www.earth-syst-sci-data-discuss.net/essd-2019-202/essd-2019-202-RC2-supplement.pdf

[Figure]

**Supplement:**

**TephraKam:** Geochemical database of glass compositions in tephra and welded tuffs from the Kamchatka volcanic arc (NW Pacific)**

Maxim V. Portnyagin1,2, Vera V. Ponomareva3, Egor A. Zelenin4, Lilia I. Bazanova3, Maria M. Pevzner4, Anastasia A. Plechova2, Aleksei N. Rogozin3, Dieter Garbe-Schönberg5

[revised manuscript text omitted]

In rare cases, tephra glasses from different volcanoes have hardly distinguishable major element composition. In this case

30 minor elements determined by EMPA (P, Cl) and trace elements by LA-ICP-MS are useful to identify source volcanoes. In Fig. 5, we illustrate this case using compositions of tephra glasses from Opala volcano (eruption OP 1356 BP) and Khangar volcano (eruption KHG6600 7490 BP). Although these tephras have very different ages, this comparison is instructive to illustrate the value of minor and trace element data to distinguish compositionally close tephras. The difference in major

5

elements is very subtle and mostly within long-term analytical uncertainty: Khangar glass has about 0.5 wt% lower Al2O3,  $\leq$ 0.2 wt% higher CaO,  $\leq$ 0.5 wt% higher K2O at given SiO2 and otherwise completely overlapping TiO2, FeO, CaO and Na2O 
[revised manuscript text omitted]

---

## Author Response (AR1)

**Response to referee comments on "TephraKam: Geochemical database of glass compositions in tephra and welded tuffs from the Kamchatka volcanic arc (NW Pacific)" by Maxim V. Portnyagin et al..**

We appreciate receiving comments on our manuscript. All the comments have been addressed during revision and helped us to improve the quality of this work.

**Bärbel Sarbas (Referee #1)**
b.sarbas@mpic.de

The term database refers to sets of related data. In the case of TephraKam, there are four individual tables (volcanoes, samples, major elements and trace elements) which are not related. Thus, the comprehensive and unique data collection should not be called database but dataset.

Thanks for this comment, but we disagree with the referee.

1) According to Wikipedia, "A database is an organized collection of data, generally stored and accessed electronically from a computer system." (https://en.wikipedia.org/wiki/Database). Our data collection satisfies this definition of a database. More importantly, the four tables in the .xlsx file (volcanoes, samples, major elements and trace elements) ARE linked (related) using unique sample number and/or volcano name. If one changes, for example, the age of sample in the "Table 2 - Samples" spreadsheet, the age will be automatically changed in the tables with major and trace elements. All interlinked cells are identified in the database description.

2) This is correct that our data collection has format, which is different from that of a true relation database (Access, SQL etc.). Our goal was to make the files as friendly for potential users as possible. Therefore, we provided the tables in MS Excel format, which is well known to most potential users. In the future, we would like to continue supporting the Excel based version, and our goal is certainly to release a web-based multi-user version of this database.

3) Depending on the editor's decision, we can change description of this data collection to as "dataset"; the wording is not of primary importance for us. However, we would prefer to keep its definition as "database" because it consists of linked data – a major feature of true database.

**Anonymous Referee #2**

I believe that this is a paper that is worthy of publication. It is the culmination of a thorough analysis of many of the tephra and weld tuff deposits produced by volcanoes in Kamchatka. This original data will prove invaluable to researchers working in the region and also on a wider global scale, where ultradistal tephras are now being found. The geochemical data is of a high quality and detailed explanations are provided on the analytical conditions of the instruments used to produce the results.
The issues I have with this paper that need to be addressed are as follows and I have also highlighted them in the attached PDF file.

We appreciate the positive comments from the reviewer and thorough reading of our manuscript. All points of criticism and proposed corrections were taken into account during revisions.

1. There is no mention of Miocene samples in the Abstract. A minor point, but needs correcting.
We do have one Miocene (5.7 Ma) sample in the collection.

The abstract has been corrected.

2. The discussion on tephra sample preparation for analysis on page 4 needs to include some reference to other studies. This has been widely discussed in the literature, but there is no reference to any other sources. Whilst the preparation of samples is thorough, it is not necessarily innovative and some acknowledgement of others is required here.

Indeed, there are many papers addressing separation of dispersed tephra particles from marine and continental sediments using chemical and physical methods, and their preparation for geochemical analyses. Preparation of proximal tephra samples in the course of this study has not required special sophisticated techniques. The technique described in section 3.1 is our original approach, which is based on the first author's 30+ years experience with studying glasses, melt inclusions and minerals from volcanic rocks. The detailed description was required by the ESSD rules, and it maybe helpful for other scientists looking for details of tephra preparation for analysis.

The following sentence has been added at the beginning of section 3.1 (p.4) to clarify the point raised by reviewer:
"Tephra samples have been prepared using our original technique developed in the past 20 years in GEOMAR (Kiel) and Vernadsky Institute (Moscow). The technique aims at uncomplicated, time- and material-effective preparation of many tephra samples for microanalytical work. The technique applies no strong heating (>50°C) and no acid leaching, which may cause chemical modification of tephra glasses (e.g. Hunt and Hill 1996). All materials used for preparation were thoroughly tested to exclude those causing chemical contamination of polished glass surface."

3. Page 4 discusses electron microprobe analysis and again there is no inclusion of any citations to other published work on this subject, other than some of the glass standards. This subject has been discussed in detail in the literature and some inclusion of this is needed.

In the revised paper, we included citations of works, which were influential for developing of our EPMA technique. The section 3.2 (p.5) begins with the following paragraph now.
"In this study, EPMA data have been obtained following published recommendations for the analytical conditions, primary and secondary reference materials, the number of analyses and other factors and procedures, which may influence the quality of EPMA data on tephra glasses and their interpretation (Froggatt, 1992; Morgan and London, 1996; 2005; Hunt and Hill, 2001; Turney et al., 2001). Particularly influential study was an intercomparison of electron microprobe data for volcanic glasses between different labs (Kuehn et al., 2011), which confirmed a high quality of our data (GEOMAR lab is #12 in paper by Kuehn et al. (2011) and allowed us to further improve our EPMA protocol."
A few more modifications have been made in all section 3.2. to include quotations of previous studies.

4. Page 6 onwards discusses LA-ICP-MS analyses and there is more inclusion of published work on this subject here, although I would expect to see more.

We agree that there were many excellent works on LA-ICP-MS tephra studies, which influenced our views and helped to achieve the best results. To acknowledge the previous studies, the beginning of the section 3.3 was modified as follows:

"In the past 25 years LA-ICP-MS became a common technique to quantify concentrations of a wide range of trace elements in tephra glasses (e.g. Westgate et al., 1994; Pearce et al., 1996; 2007; 2014; Tomlinson et al., 2010; Kimura and Chang, 2012; Maruyama et al., 2016;

Lowe et al., 2017). The LA-ICP-MS technique adopted for this study and its development were based on the principle results and recommendations from the previous works, and our own experimental results. All the trace element analyses were obtained at the Institute of Geosciences, CAU Kiel, Germany. Conditions of analysis are summarized in TephraKam Table 1d (Portnyagin et al., 2019)."
Some additional references were included in the following text.

5. I am not suggesting that this is done for the publication of this paper, but it would be good to see this data housed in a proper relational database. This would allow more complex searching than is currently possible with the Excel spreadsheets. Some mention of this as at least a possibility would be good.

We addressed this issue in the response to the reviewer #1. Although in our opinion, Excel allows very complex searching and filtering the data covering all needs for tephra studies, we do plan to release a web-based version of this database in near future. A note about this plan has been made in section 5 (p. 14).

6. Although not badly written, there are a lot of minor English issues with the text. I have tried to highlight these in the PDF file (Supplement) and I hope the authors find this useful. I do appreciate the difficulties in writing a paper in another language and I feel that they have done a good job, but the quality of the writing needs to be more consistent. I would hope the editors/publishers would also contribute to this. Please also note the supplement to this comment: https://www.earth-syst-sci-data-discuss.net/essd-2019-202/essd-2019-202-RC2-supplement.pdf

We appreciate very much these corrections and do find them useful. We have read the manuscript thoroughly and made a number of corrections to the style and grammar. We hope that the writing meets the journal requirements now.

References were formatted following ESSD rules in the revised version.

In addition, we performed the following important work on the data files:
1) A few misprints were corrected in the sample description table.
2) The analytical conditions were verified and corrected in few parts.
3) Major element contents normalized to 100% were corrected to account for halogen substitution of oxygen.
4) The link to updated database (version 2020.01.30) was placed on ResearchGate under the same address (DOI) as before.

[revised manuscript text omitted]